# Interval Cancer Rate and Diagnostic Performance of Fecal Immunochemical Test According to Family History of Colorectal Cancer

**DOI:** 10.3390/jcm9103302

**Published:** 2020-10-14

**Authors:** Yoon Suk Jung, Jinhee Lee, Hye Ah Lee, Chang Mo Moon

**Affiliations:** 1Division of Gastroenterology, Department of Internal Medicine, Kangbuk Samsung Hospital, Sungkyunkwan University School of Medicine, Seoul 03181, Korea; ys810.jung@samsung.com; 2Department of Endocrinology and Metabolism, Ajou University School of Medicine, Suwon 16499, Korea; marie0715@naver.com; 3Clinical Trial Center, Ewha Womans University Mokdong Hospital, Seoul 07985, Korea; hyeah@ewha.ac.kr; 4Department of Internal Medicine, College of Medicine, Ewha Womans University, Seoul 07985, Korea; 5Inflammation-Cancer Microenvironment Research Center, College of Medicine, Ewha Womans University, Seoul 07985, Korea

**Keywords:** family history of colorectal cancer, fecal immunochemical test, colorectal cancer

## Abstract

Background: The potential role of the fecal immunochemical test (FIT) in individuals with a family history of colorectal cancer (CRC) remains unclear. We assessed interval cancer rate (ICR) after the FIT and FIT diagnostic performance according to family history of CRC. Methods: Using the Korean National Cancer Screening Program Database, we collected data on subjects who underwent the FIT between 2009 and 2011. The interval cancer rate (ICR) was defined as the number of subjects diagnosed with CRC within 1 year after the FIT per 1000 subjects with negative FIT results. Results: Of 5,643,438 subjects, 224,178 (3.97%) had a family history of CRC. FIT positivity rate (6.4% vs. 5.9%; adjusted relative risk (aRR) 1.11; 95% confidence interval (CI) 1.09–1.13) and ICR (1.4 vs. 1.1; aRR 1.43 (95% CI 1.27–1.60)) were higher in these subjects than in those with no such history. These results were the same regardless of whether subjects had undergone colonoscopy within the last 5 years before the FIT. However, the diagnostic performance of the FIT for CRC, as measured using the area under the operating characteristic curve, was similar between subjects without a family history and those with one (85.5% and 84.6%, respectively; *p* = 0.259). Conclusion: the FIT was 1.4 times more likely to miss CRC in subjects with a family history than in those without (aRR 1.43 for ICR), although its diagnostic performance was similar between the two groups. Our results suggest that for individuals with a family history of CRC, colonoscopy should be preferred over FIT for both screening and surveillance.

## 1. Introduction

Among all cancers worldwide, colorectal cancer (CRC) is the second leading cause of mortality and has the fourth highest incidence [1]. A family history of CRC is a strong and well-known risk factor for CRC [2,3]. Individuals with a family history of CRC have been reported to be two to four times as likely to develop CRC as those without [2,4]. Accordingly, existing guidelines on CRC screening generally recommend colonoscopy over the fecal immunochemical test (FIT) for subjects with a strong family history, such as those with a first-degree relative (FDR) diagnosed with CRC at a relatively young age and those with two or more FDRs diagnosed with CRC [5,6,7]. For instance, the United States’ guidelines recommend colonoscopy every 5 years for those with two FDRs diagnosed with CRC or one FDR diagnosed at under 60 years of age, while for individuals with one FDR diagnosed with CRC at age 60 or above, screening recommendations are the same as those for average-risk individuals (either annual FIT or colonoscopy every 10 years) [5]. However, these recommendations are based on very low-quality evidence.

Some clinicians suggest considering the FIT as a primary screening strategy for individuals with a family history of CRC, citing concerns about poor adherence to colonoscopy [8]. However, it remains unclear whether screening via the FIT instead of colonoscopy is useful for this high-risk group. Several previous studies have evaluated the accuracy of the FIT for detecting CRC in these individuals [9,10,11]. However, the number of subjects involved in these studies was too small to provide reliable results, and they did not directly compare FIT accuracy between individuals with a family history of CRC and those without one. It is still questionable whether the FIT can serve the same role in screening high-risk individuals with a family history of CRC as it does for average-risk individuals without a family history of CRC. Given these problems, we sought to clarify the potential role of the FIT for individuals with a family history of CRC by analyzing data from a large-scale, nationwide, population-based CRC screening program. We compared FIT positivity rate, interval cancer rate (ICR), and FIT diagnostic performance between individuals with a family history of CRC and those without one.

## 2. Materials and Methods

### 2.1. Study Population

The Korea National Cancer Screening Program (NCSP) provides a single annual FIT as an initial CRC screening examination for all Koreans over 50 years of age and those with a positive FIT are offered a colonoscopy as a second examination. Participants were notified of the FIT results (reported as “positive” or “negative”) by mail within 15 days. Additionally, participants who had a positive FIT result were contacted by telephone by the medical staff, informed of the positive result, and offered an appointment date for follow-up colonoscopy. All of these examinations were performed free of charge at a clinic or hospital designated as a CRC screening unit by the National Health Insurance Corporation (NHIC). To be designated as a CRC screening unit, a clinic or hospital must have colonoscopy equipment and at least one fulltime medical doctor [12]. The population for the current study comprised men and women aged 50 years or older who received FITs through the NCSP between 1 January 2009, and 31 December 2011. Data were extracted from the National Health Information Database (NHID) of the National Health Insurance Service (NHIS), which runs the NCSP.

The NHIS–NHID is encrypted and does not contain personal identifiers. This study was approved by the institutional review board of Ewha Womans University Mokdong Hospital (IRB No. 2020-06-028).

### 2.2. Fecal Immunochemical Test

Because the NCSP provides a single annual FIT, FIT results were processed using only one sampling, either by qualitative or quantitative methods. For those with two or more FITs during the study period, we included only the initial FIT results. For the qualitative FIT, SD Bioline FOB kits (SD, Co., Korea), with a cutoff point of 30 ng/mL; ASAN Easy Test FOB kits (Asan Pharm, Co., Korea), with a cutoff point of 50 ng/mL; FOB test kits (Humasis, Co., Korea), with a cutoff point of 50 ng/mL; and OC-Hemocatch Light kits (Eiken Chemical, Co., Tokyo, Japan), with a cutoff point of 50 ng/mL, were applied. For the quantitative FIT, Medex HM-JACK kits (Kyowa Chemical Industry, Co., Kagawa, Japan), with a cutoff point of 30 ng/mL; Hemo Tech NS-1000 kits (Alfresa Pharma, Co., Osaka, Japan), with a cutoff point of 40 ng/mL; and OC-Sensor DIANA kits (Eiken Chemical, Co.), with cutoff point of 100 ng/mL, were used [12,13].

### 2.3. Definition of CRC Status and Variables of Interest

The age, sex, date of screening examination, FIT results, and International Classification of Diseases 10th Revision (ICD-10) codes were obtained from the NHIS–NHID. The CRC status of included subjects was defined by using the database; subjects coded according both the ICD-10 (C18–C21, D01.0–D01.3) and the Korean national cancer registration program were recorded as having CRC. CRC incidence was defined as the rate of CRC diagnosis within 1 year after the FIT.

Data on family history of CRC were collected on the basis of replies to a self-administered questionnaire. To receive the FIT provided by the NCSP, examinees are obliged to fill out a questionnaire that includes questions regarding family history of CRC. Subjects were recorded as having a family history of CRC if they had any FDRs (parents, siblings, or children) who had been diagnosed with CRC, regardless of age at diagnosis.

Whether subjects had undergone a colonoscopy prior to the FIT was determined by checking for a corresponding prescription code. We checked all subjects’ records from 1 January 2002, to 31 December 2011, for codes for colonoscopy, colonoscopic polypectomy, colonoscopic mucosal resection, or colonoscopic submucosal resection. The interval since colonoscopy was defined as the interval between the date of the FIT and the most recent colonoscopy before the FIT.

### 2.4. Statistical Analysis

Baseline characteristics were compared between subjects with and without a family history of CRC using the χ^2^ test for categorical variables and the Student’s *t*-test for continuous variables.

We also compared FIT positivity rate, cancer detection rate (CDR), ICR, and four parameters of FIT diagnostic performance between subjects with and without a family history of CRC. FIT positivity rate was defined as the proportion of all subjects whose FIT results were positive. CDR was defined as the number of subjects diagnosed with CRC within 1 year after the FIT per 1000 subjects who underwent the FIT, regardless of results. ICR was defined as the number of subjects diagnosed with CRC within 1 year after the FIT per 1000 subjects with negative FIT results [13]. Sensitivity was defined as the number of true positives divided by the total number of subjects diagnosed with CRC within 1 year after the FIT. Specificity was defined as the number of true negatives divided by the total number of subjects not diagnosed with CRC within 1 year after the FIT. Positive predictive value (PPV) was defined as the proportion of subjects with positive FIT results who were diagnosed with CRC within 1 year after the FIT, while negative predictive value (NPV) was defined as the proportion of subjects with negative FIT results who were not diagnosed with CRC within 1 year after the FIT. For all seven of these variables, we conducted age- and sex-adjusted Poisson regression modeling with the log link function to assess potential differences according to family history of CRC. This modeling method gave us age- and sex-adjusted relative risk values (adjusted relative risk, aRR) and their corresponding 95% confidence intervals (CIs). The area under the receiver operating characteristic curve (AUROC) was also calculated for subjects with and without a family history of CRC, and the χ^2^ test for homogeneity of areas was used to determine whether these groups showed a statistically significant difference in the diagnostic accuracy of the FIT.

All reported *p*-values were two-sided, and *p* < 0.05 was considered statistically significant. All data analyses were performed using the SAS software, version 9.4 (SAS Institute, Cary, NC, USA).

## 3. Results

### 3.1. Participants and Baseline Characteristics

Of the 6,343,240 participants who underwent the FIT for CRC screening through the NCSP between 2009 and 2011, we excluded those with previous diagnoses of cancer (including CRC) (*n* = 378,795) or inflammatory bowel disease (IBD) (*n* = 22,071). Another 194,568 subjects were excluded because of the absence of data on screening date, age, sex, or family history of CRC; an additional 44,586 were excluded due to un-certificated quality assurance at the screening units. Subjects were grouped according to whether they had undergone a colonoscopy within the last 5 years before their FIT, and those who had one during the last 180 days before the FIT (*n* = 58,782) were excluded as this interval was short enough that the FIT and colonoscopy could be considered to have been performed at approximately the same time. Ultimately, a total of 5,643,438 participants who underwent the FIT in the study period were included in the analysis (Figure 1). The history of cancer, including CRC (C, D01.0–D01.3) and IBD (K50, K51), was confirmed through the presence of the corresponding diagnostic code before the FIT.

Of the 5,643,438 participants, 224,178 (3.97%) had a family history of CRC. Baseline characteristics of the study population are shown in Table 1. The mean age of the study population was 60.6 years (SD 8.2 years), and 43.7% were men. The subjects without a family history of CRC included more men and had a greater mean age than those who had such a history. The age ranges for those with and without family history of CRC were 50–100 years and 50–108 years, respectively. The proportion of subjects who had undergone colonoscopy within the last 5 years before the FIT was higher among those with a family history of CRC than among those without (20.0% vs. 15.2%, *p* < 0.001).

With regard to FIT analysis methods, the qualitative and quantitative analysis accounted for 71.1% and 28.9%, respectively, in participants with family history of CRC, while qualitative and quantitative analysis accounted for 68.6% and 31.4%, respectively, in those without family history of CRC (Appendix A).

### 3.2. FIT Positivity Rate, CDR, and ICR

Table 2 shows the results obtained when FIT positivity rate, CDR, and ICR were compared according to family history of CRC. As expected, subjects with a family history of CRC had a significantly higher FIT positivity rate (6.4% vs. 5.9%, aRR 1.11 (95% CI 1.09–1.13)) and CDR (3.8 per 1000 subjects vs. 2.9 per 1000, aRR 1.43 (95% CI 1.33–1.53)) than those without. ICR was also significantly higher in subjects with a family history of CRC (1.4 per 1000 subjects with negative FIT results vs. 1.1 per 1000, aRR 1.43 (95% CI 1.27–1.60)). These differences were observed both among subjects who had undergone colonoscopy within the last 5 years before the FIT and among those who had not.

### 3.3. Diagnostic Performance of FIT

Table 3 shows the results obtained when the sensitivity, specificity, PPV, and NPV of the FIT were compared according to family history of CRC. Sensitivity was similar between subjects with and without family history of CRC (64.4% vs. 64.1%, aRR 1.01 (95% CI 0.92–1.10)). PPV was higher in subjects with a family history of CRC than in those without (3.8% vs. 3.1%, aRR 1.29 (95% CI 1.18–1.41)), whereas NPV was similar between the two groups (99.9% vs. 99.9%, aRR 1.00 (95% CI 0.995–1.004)). The results were similar among subjects who had not undergone colonoscopy within the last 5 years before the FIT: PPV was higher in subjects with a family history of CRC (4.4% vs. 3.5%, aRR 1.33 (95% CI 1.22–1.45)), while sensitivity and NPV were similar between the two groups. However, among subjects who had undergone colonoscopy during this period, family history of CRC was not associated with significant differences in any of these parameters.

The AUROC for the diagnostic accuracy of the FIT was 85.5% (95% CI 85.1%–85.8%) in subjects without a family history of CRC and 84.6% (95% CI 83.0%–86.1%) in those with such a history (Figure 2A). There was no significant difference between the two groups (*p* = 0.259). For subjects who had not undergone colonoscopy within the last 5 years before the FIT, the AUROC was 86.0% for those with no family history of CRC and 85.0% for those with such a history (Figure 2B), with no significant difference (*p* = 0.273); similarly, for subjects who had undergone colonoscopy within the last 5 years before the FIT, the AUROC was 77.6% for those with no family history of CRC and 77.5% for those with such a history (Figure 2C), with no significant difference (*p* = 0.967). Although family history of CRC was not significant with regard to the AUROC, personal history of colonoscopy was; the AUROC was significantly higher for those who had not undergone colonoscopy within the last 5 years before the FIT than for those who had (no family history of CRC: 86.0% vs. 77.6%, *p* < 0.001; with family history of CRC: 85.0% vs. 77.5%, *p* = 0.038).

## 4. Discussion

In this Korean population-based study, we found that the FIT positivity rate, CDR, and ICR were higher among subjects with a family history of CRC than among those without one. However, there was no difference between the two groups with regard to the diagnostic performance of the FIT as measured by the AUROC. The higher ICR in subjects with a family history of CRC is a particularly important finding; the ICR was about 1.4 times higher in these subjects (aRR 1.43) than in those without such a history. Because the ICR was calculated as the proportion of subjects with negative FIT results who were later diagnosed with CRC, it can be regarded as the rate at which the FIT failed to detect CRC. In other words, we found that the FIT was 1.4 times more likely to miss CRC in subjects with a family history of CRC. This may be because the adenoma–carcinoma sequence may progress more rapidly in individuals with a family history of CRC [14,15], and the FIT has limited effectiveness for detecting advanced adenoma and early-stage CRC [16,17]. Our findings suggest that colonoscopy, rather than the FIT, is the best option for CRC screening in high-risk individuals with a family history of CRC. Such a history was associated with a higher ICR both in subjects who underwent colonoscopy, including colonoscopic polypectomy, in the last 5 years before the FIT and in those who did not; therefore, our results suggest that colonoscopy may be preferable to the FIT in these subjects even for surveillance after colonoscopy or polypectomy.

However, a possible drawback of recommending colonoscopy to individuals with a family history of CRC is that the adherence rate may be low. Indeed, there have been studies showing low adherence to colonoscopy screening among FDRs of CRC patients (38–43%) [18,19]. To improve the effectiveness of CRC screening in these individuals, strategies to improve participation rates should be developed, such as educating the public regarding the importance of colonoscopy.

Similarly to us, the authors of a Hong Kong study involving 4539 Chinese patients recommended colonoscopy as the preferred choice for CRC screening in individuals with a family history of CRC [20]. This recommendation was based on their findings that the FIT was more likely to miss advanced neoplasia and CRC in individuals with a family history of CRC; among 513 FIT-negative subjects with a family history of CRC, advanced neoplasia was found in 24 individuals (4.7%) and CRC in three (0.6%) [20]. In contrast, a prospective randomized trial of 1918 FDRs of patients with CRC found that repeated FIT screening (once per year for 3 years) was equally effective as one-time colonoscopy in detecting advanced neoplasia and successfully detected all CRCs [11]. However, a one-time FIT is very likely to be inferior to a one-time colonoscopy. Moreover, in actual clinical practice, the compliance rate for three consecutive annual FITs is likely to be low. When the results of these previous studies are considered together with ours, they indicate that colonoscopy should be the first-line screening option, with the FIT being a second-line option to be applied only if colonoscopy capacity is very limited or the patient refuses colonoscopy.

As expected, the FIT positivity rate and CDR were higher in subjects with a family history of CRC than in those without, supporting the view that a family history of CRC is a strong risk factor for CRC. On the other hand, there were no significant differences between these groups with regard to the AUROC, FIT sensitivity, or FIT NPV. In other words, the diagnostic performance of the FIT was similar in both of these groups. A recent meta-analysis of 12 studies with a combined population of 6204 found FIT to have high overall accuracy for CRC detection in individuals with a family history of CRC (86% sensitivity and 91% specificity); likewise, it concluded that FIT accuracy in these individuals is reasonable and similar to that observed for average-risk persons [21]. However, the conclusions that could be drawn from this meta-analysis were limited owing to the high heterogeneity between studies, low prevalence of CRC, small sample size, and large CIs. A study from Hong Kong also reported that there was no significant difference in FIT sensitivity for CRC between subjects without and with a family history of CRC, although it was somewhat higher in those without (61.1% (*n* = 11/18) vs. 25.0% (*n* = 1/4), *p* = 0.190) [20]. However, that study used a qualitative FIT, not a quantitative test, and the number of patients with CRC was too small for the reliable assessment of sensitivity. Our study, with a very large sample size drawn from a nationwide population-based screening program, can provide more reliable information on the performance of the FIT in individuals with a family history of CRC.

In the present study, we found that among subjects who did not undergo colonoscopy within the last 5 years before the FIT, PPV was higher in those with a family history of CRC than in those without, whereas among those who did undergo colonoscopy during this period, there were no significant differences in PPV with regard to family history of CRC. This may be because of precancerous lesions, including advanced adenomas, being removed at the time of the colonoscopy.

Interestingly, the diagnostic accuracy of the FIT as measured by the AUROC showed significant differences depending on history of colonoscopy. It was lower in subjects who had undergone colonoscopy within the last 5 years before the FIT than in those who had not, both among those with a family history of CRC and among those without. This suggests that the FIT performance is reduced for subjects with a history of colonoscopy, regardless of family history of CRC, implying that the interim FIT between colonoscopic examinations may not be helpful even for individuals with a family history of CRC.

In the present study, the sensitivity of the FIT for CRC detection was 64%, which was lower compared to the results of previous studies. A meta-analysis of 19 studies demonstrated that the pooled sensitivity of the FIT for CRC was 79% (95% CI, 0.69–0.86) in asymptomatic average-risk adults [22]. One of the reasons for the lower sensitivity of the FIT in our study may be because we considered only one-time FIT results. It may also be because the brands of FIT device and cutoff values for a positive test result vary from study to study. Therefore, it may be difficult to generalize our findings to other countries where a higher sensitivity of the FIT has been reported. Worldwide studies are needed to validate our results.

This is the largest study to date evaluating the ICR and FIT performance with regard to CRC in individuals with a family history of CRC. Nevertheless, it has several limitations. First, data on family history of CRC were self-reported and therefore might have been underreported. However, the proportion of NHIS–NHID entries missing these data was very low (3%). Second, the number of FDRs diagnosed with CRC and their age at diagnosis could not be analyzed because of the limitations of the NHIS–NHID. Third, since information on the date and results of colonoscopy was insufficient in the NHIS-NHID, the cancer rate missed by colonoscopy could not be analyzed. Additionally, for the same reason, the FIT performance for advanced adenoma could not be assessed. Fourth, because the Korean government does not designate a single instrument for FITs provided through the NCSP, screening involved various brands of FIT device with varying cutoff points. Fifth, the brand name of the FIT kit was not included in the NHIS-NHID and the cutoff level was not included in the case of qualitative analysis. In addition, uncertain cutoff levels existed in some cases with quantitative analysis. Accordingly, we could not compare the cutoff level of the FIT according to family history. Sixth, because only subjects who participated in FIT screening were analyzed, there may be some degree of selection bias. During the study period, the participation rate of CRC screening (FIT) was low, ranging from 27–33% (Appendix A) [23]. Seventh, there was a significant difference in the history of colonoscopy within past 5 years between individuals without and with a family history of CRC. A higher percentage of colonoscopy history in individuals with a family history than in those without a family history may have acted as a potential biasing factor in lowering the positive rate and diagnostic yield of the FIT in those with a family history. Lastly, subjects’ diagnoses of CRC were not confirmed by colonoscopy. However, we defined subjects as having CRC if they were coded as such according to both the ICD-10 and the Korean national cancer registration program. The Korean government, through the NHIS, manages a registration program for all cancer patients that subsidizes their medical expenses. A cancer diagnosis is unlikely to be missed to ensure access to these financial benefits. In addition, registration requires confirmation of the cancer diagnosis using strict criteria based on histological examination. Therefore, it is unlikely that the NHIS–NHID incorrectly identified CRC in individuals who did not have it.

In conclusion, certain parameters of the FIT performance with regard to CRC diagnosis, such as sensitivity and the AUROC, were similar in subjects with and without a family history of CRC. However, the FIT was 1.4 times more likely to miss CRC in subjects with a family history of CRC than in those without. A higher ICR was observed in subjects with a family history of CRC, both in those who had undergone colonoscopy within the last 5 years before the FIT and in those who had not. Our findings suggest that for individuals with a family history of CRC, colonoscopy should be preferred over the FIT for both screening and surveillance.

## Figures and Tables

**Figure 1 jcm-09-03302-f001:**
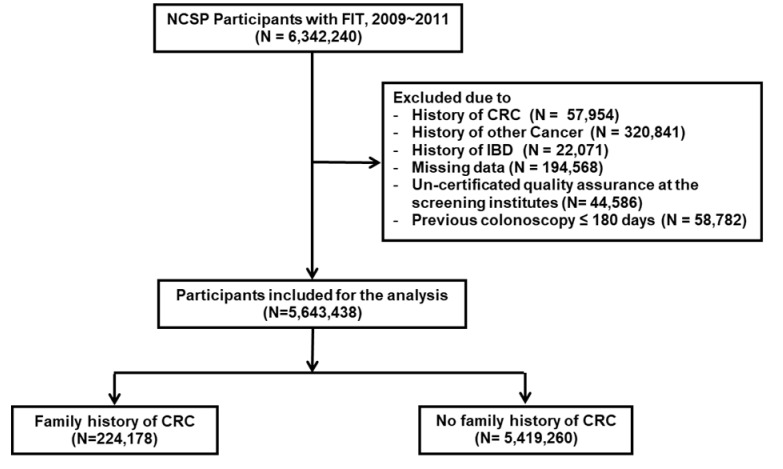
Flowchart of study population selection. FIT, fecal immunochemical test; CRC, colorectal cancer; IBD, inflammatory bowel disease.

**Figure 2 jcm-09-03302-f002:**
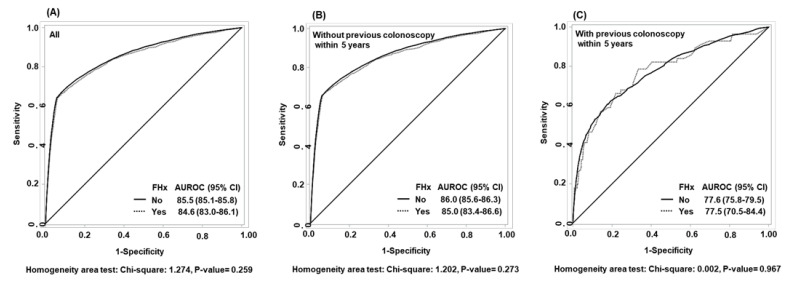
Receiver operating characteristic curves for the fecal immunochemical test according to family history of colorectal cancer in all subjects (**A**), those with no history of colonoscopy in the last 5 years before testing (**B**), and those who underwent a colonoscopy in the last 5 years before testing (**C**). FHx, family history; AUROC, area under the receiver operating characteristic curve; CI, confidence interval.

**Table 1 jcm-09-03302-t001:** Baseline characteristics according to family history of CRC.

	Total N = 5,643,438	Family History of CRC *n* = 224,178	No Family History of CRC *n* = 5,419,260	*p*-value
Sex				
Male	2,465,771 (43.7)	95,298 (42.5)	2,370,473 (43.7)	<0.001
Female	3,177,667 (56.3)	128,880 (57.5)	3,048,787 (56.3)	
Age (years)	60.6 ± 8.2	59.6 ± 7.8	60.7 ± 8.2	<0.001
50–59	2,796,297 (49.5)	123,579 (55.1)	2,672,718 (49.3)	<0.001
60–69	1,878,505 (33.3)	70,750 (31.6)	1,807,755 (33.4)	
70–79	857,530 (15.2)	26,805 (12.0)	830,725 (15.3)	
≥80	111,106 (2.0)	3,044 (1.4)	108,062 (2.0)	
History of colonoscopy within past 5 years				
No	4,773,868 (84.6)	179,259 (80.0)	4,594,609 (84.8)	<0.001
Yes	869,570 (15.4)	44,919 (20.0)	824,651 (15.2)	
History of polypectomy	186,437 (3.3)	9,887 (4.4)	176,550 (3.3)	<0.001
FIT positive	334,195 (5.9)	14,385 (6.4)	319,810 (5.9)	<0.001
CRC detection *	16,363 (0.29)	845 (0.38)	15,518 (0.29)	<0.001
Interval cancer **	5,865/5,309,243 (0.11)	301/209,793 (0.14)	5,564/5,099,450 (0.11)	<0.001

Values are presented as number (%) or mean ± SD. FIT, fecal immunochemical test; CRC, colorectal cancer. * The number (%) of subjects diagnosed with CRC within 1 year after the FIT, regardless of FIT results. ** The number (%) of subjects diagnosed with CRC within 1 year after the FIT among subjects with negative FIT results.

**Table 2 jcm-09-03302-t002:** FIT positive rate, cancer detection rate, and interval cancer rate in participants with and without a family history of CRC.

	FIT Positive Rate			Cancer Detection Rate			Interval Cancer Rate		
	% (95% CI)	aRR (95% CI)	*p*-value	Per 1000 * (95% CI)	aRR (95% CI)	*p*-value	Per 1000 ** (95% CI)	aRR (95% CI)	*p*-value
All participants									
FHx of CRC									
No	5.9 (5.9–5.9)	1.00 (ref)		2.9 (2.8–2.9)	1.00 (ref)		1.1 (1.1–1.1)	1.00 (ref)	
Yes	6.4 (6.3–6.5)	1.11 (1.09–1.13)	<0.001	3.8 (3.5–4.0)	1.43 (1.33–1.53)	<0.001	1.4 (1.3–1.6)	1.43 (1.27–1.60)	<0.001
Without previous colonoscopy ≤5 years									
FHx of CRC									
No	6.0 (5.9–6.0)	1.00 (ref)		3.2 (3.2–3.3)	1.00 (ref)		1.2 (1.2–1.2)	1.00 (ref)	
Yes	6.6 (6.5–6.7)	1.12 (1.10–1.14)	<0.001	4.4 (4.1–4.7)	1.48 (1.38–1.59)	<0.001	1.6 (1.4–1.8)	1.46 (1.29–1.65)	<0.001
With previous colonoscopy ≤5 years									
FHx of CRC									
No	5.5 (5.5–5.6)	1.00 (ref)		0.9 (0.9–1.0)	1.00 (ref)		0.6 (0.5–0.6)	1.00 (ref)	
Yes	5.8 (5.6–6.0)	1.07 (1.03–1.11)	<0.001	1.2 (0.9–1.6)	1.50 (1.15–1.97)	0.003	0.8 (0.5–1.0)	1.51 (1.06–2.15)	0.023

Relative risk was adjusted for age and sex. * The number of subjects diagnosed with CRC within 1 year after the FIT per 1000 subjects, regardless of FIT results. ** The number of subjects diagnosed with CRC within 1 year after the FIT per 1000 subjects with negative FIT results. FIT, fecal immunochemical test; CRC, colorectal cancer; aRR, adjusted relative risk; CI, confidence interval; FHx, family history; ref, reference.

**Table 3 jcm-09-03302-t003:** Diagnostic performance of the FIT in participants with and without a family history of CRC.

	Sensitivity			Specificity			PPV			NPV		
	% (95% CI)	aRR (95% CI)	*p-*value	% (95% CI)	aRR (95% CI)	*p-*value	% (95% CI)	aRR	*p-*value	% (95% CI)	aRR	*p-*value
Total												
FHx of CRC												
No	64.1 (63.4–64.9)	1.00 (ref)		94.3 (94.2–94.3)	1.00 (ref)		3.1 (3.1–3.2)	1.00 (ref)		99.9 (99.9–99.9)	1.00 (ref)	
Yes	64.4 (61.2–67.6)	1.01 (0.92–1.10)	0.903	93.8 (93.7–93.9)	0.994 (0.990–0.999)	0.008	3.8 (3.5–4.1)	1.29 (1.18–1.41)	<0.001	99.9 (99.8–99.9)	1.000 (0.995–1.004)	0.849
Without previous colonoscopy ≤5 years												
FHx of CRC												
No	65.3 (64.6–66.1)	1.00 (ref)		94.2 (94.2–94.2)	1.00 (ref)		3.5 (3.5–3.6)	1.00 (ref)		99.9 (99.9–99.9)	1.00 (ref)	
Yes	66.0 (62.7–69.3)	1.01 (0.93–1.11)	0.785	93.7 (93.6–93.8)	0.993 (0.989–0.998)	0.008	4.4 (4.1–4.8)	1.33 (1.22–1.45)	<0.001	99.8 (99.8–99.9)	1.000 (0.995–1.004)	0.841
With previous colonoscopy ≤5 years												
FHx of CRC												
No	41.4 (37.9–44.9)	1.00 (ref)		94.5 (94.4–94.5)	1.00 (ref)		0.7 (0.6–0.8)	1.00 (ref)		99.9 (99.9–99.9)	1.00 (ref)	
Yes	41.1 (28.2–54.0)	1.00 (0.66–1.53)	0.990	94.2 (94.0–94.4)	0.996 (0.986–1.006)	0.435	0.9 (0.5–1.2)	1.41 (0.92–2.15)	0.114	99.9 (99.9–99.9)	1.00 (0.99–1.01)	0.958

Relative risk was adjusted for age and sex. FIT, fecal immunochemical test; CRC, colorectal cancer; PPV, positive predictive value; NPV, negative predictive value; aRR, adjusted relative risk; CI, confidence interval; FHx, family history; ref, reference.

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
