# Peer review of "Interval Cancer Rate and Diagnostic Performance of Fecal Immunochemical Test According to Family History of Colorectal Cancer"

_jcm, 2020, doi:10.3390/jcm9103302_

Round 1
Reviewer 1 Report
None
Author Response
Thank you for your review.
Reviewer 2 Report
Dear Editor,
Thank you for the opportunity to review the manuscript titled “Interval cancer rate and diagnostic performance of fecal immunochemical test according to family history of colorectal cancer”. The authors presented analysis of data from the Korea National Cancer Screening Program on 5,643,438 participants who underwent faecal immunochemical test as part of colorectal cancer (CRC) screening. The authors compared the data on diagnostic performance of FIT between individuals with and without family history of CRC. The main results showed a slightly higher proportion of positive results among individuals with family history of CRC. FIT positivity rate was 6.4% in individuals with a positive family history and 5.9% in individuals with no family history of CEC (adjusted relative risk [aRR] 1.11; 95% confidence interval [CI] 1.09–1.13). There were no significant differences between study groups in terms of sensitivity, specificity, negative and positive predictive values. The major finding of this study is the rate of interval cancer diagnosed 1.4 times higher in subjects with family history of CRC (aRR 1.43) than in those without.
I believe that this is a valuable paper which could add to the literature, however, I have several serious considerations regarding this study. Two main objections include the following issues:
- the authors report 4 different cut-off levels of haemoglobin in stool for a positive result ranging from 30 ng/ml to 100 ng/ml. The distribution of the cut-off levels among study population is not reported and compared between individuals with and without family history for CRC. This must be clarified by the authors as it significantly influences the interpretation of the results,
- the authors report interval CRCs diagnosed as all the CRC diagnoses within a year from performing FIT. They do not provide information what was the proportion of the CRCs diagnosed after positive FIT and subsequent colonoscopy, therefore it is unclear how many cancers were detected by FIT but missed by colonoscopy. In these cases, diagnosed CRCs should be diagnosed as interval CRC after some period of time (usually 6 months) from colonoscopy.
I strongly believe that these should be addressed before the potential publication.
Other comments:
Abstract:
- Results: I would suggest to report the number of interval CRCs detected in both study groups
Methods:
- Page 2, Study population section: the authors should provide more detailed description of the screening process in Korea. Age range for individuals with and without family history of CRC should be reported.
- Description of the study population should also include how the history of CRC is determined (i.e. how information on family history is obtained)
- A description of how the individuals eligible for screening are identified should be included
- Does The Korea National Cancer Screening Program keep data in a central database?
- Page 2, lines 72-79: the numbers reported by the authors should be moved to the results section
- Page 2, line 79 – please specify if the reported final number of individuals in the study population reflects the number of individuals who underwent FIT in the study period (although it is clearly shown on the Figure 1, it would be valuable for a reader to report it in the text)
- Page 3, lines 87-97 – the authors report 4 different cut-off levels of haemoglobin in stool for a positive results ranging from 30 ng/ml to 100 ng/ml. The distribution of the cut-off levels among study population should be reported and compared between individuals with and without family history for CRC, as this significantly influences the interpretation of the results
- The rate of 3.97% for having a positive family history of CRC is relatively low.
- There was a significant difference between the proportions of individuals who underwent colonoscopy in the last 5 years in the study groups. This should be discussed in the discussion section as a potential biasing factor lowering the positivity rate and diagnostic yield among individuals with a positive family history. Both individuals with and without family history of CRC who underwent colonoscopy within the last 5 years had significantly lower proportion of positive test results.
- The authors should also discuss the possibility that interval cancers are a consequence of inadequate colonoscopy quality performed as a workup for a positive FIT. The proportion of interval cancers diagnosed after negative and positive FITs should be reported. The data on colonoscopy quality requirements in Korea should also be reported.
- Page 7, lines 4-6 – the statement that ‘FIT specificity was lower for subjects with a family history of CRC than those without’ is too strong, as the significance of this difference is very low (93.8% vs. 94.3%, aRR 0.994 [95% CI 0.990–0.999]). This comment also applies to lines 9-10 on the same page.
- The proportion of individuals who had previous screening with a positive result (i.e. a polyp which was removed) should be reported in two study groups.
- The discussion lacks a limitations section. Limitations of this study should be discussed to provide the reader better insight to the presented results.
Author Response
We have attached our point-by-point replies to the reviewers’ specific comments as word file.
Comments and Suggestions for Authors
Dear Editor,
Thank you for the opportunity to review the manuscript titled “Interval cancer rate and diagnostic performance of fecal immunochemical test according to family history of colorectal cancer”. The authors presented analysis of data from the Korea National Cancer Screening Program on 5,643,438 participants who underwent faecal immunochemical test as part of colorectal cancer (CRC) screening. The authors compared the data on diagnostic performance of FIT between individuals with and without family history of CRC. The main results showed a slightly higher proportion of positive results among individuals with family history of CRC. FIT positivity rate was 6.4% in individuals with a positive family history and 5.9% in individuals with no family history of CEC (adjusted relative risk [aRR] 1.11; 95% confidence interval [CI] 1.09–1.13). There were no significant differences between study groups in terms of sensitivity, specificity, negative and positive predictive values. The major finding of this study is the rate of interval cancer diagnosed 1.4 times higher in subjects with family history of CRC (aRR 1.43) than in those without.
I believe that this is a valuable paper which could add to the literature, however, I have several serious considerations regarding this study. Two main objections include the following issues:
the authors report 4 different cut-off levels of haemoglobin in stool for a positive result ranging from 30 ng/ml to 100 ng/ml. The distribution of the cut-off levels among study population is not reported and compared between individuals with and without family history for CRC. This must be clarified by the authors as it significantly influences the interpretation of the results,
the authors report interval CRCs diagnosed as all the CRC diagnoses within a year from performing FIT. They do not provide information what was the proportion of the CRCs diagnosed after positive FIT and subsequent colonoscopy, therefore it is unclear how many cancers were detected by FIT but missed by colonoscopy. In these cases, diagnosed CRCs should be diagnosed as interval CRC after some period of time (usually 6 months) from colonoscopy.
I strongly believe that these should be addressed before the potential publication.
Reply: Thank you for your comments. As the reviewer pointed out, we investigated FIT analysis methods in more detail. As a result, qualitative and quantitative analysis accounted for 71.1% and 28.9%, respectively, in participants with family history of CRC, while qualitative and quantitative analysis accounted for 68.6% and 31.4%, respectively, in those without family history of CRC. In the data source (National Health Information Database [NHID] of the National Health Insurance Service [NHIS]) we analyzed, the brand name of each FIT kit was not included and the cutoff level was also not included in the case of qualitative analysis. Because qualitative analyses without cutoff level information occupied a very large proportion of study subjects and uncertain cutoff levels existed in some cases with quantitative analysis, we could not compare the distribution of the cutoff levels according to family history. We have added this limitation in the Discussion section as follows: “Fifth, the brand name of FIT kit was not included in the NHIS-NHID and the cutoff level was not included in the case of qualitative analysis. In addition, uncertain cutoff levels existed in some cases with quantitative analysis. Accordingly, we could not compare the cutoff level of FIT according to family history.”
|
Family History of CRC N=224,178 |
No Family History of CRC N = 5,419,260 |
Qualitative |
159,398 (71.1) |
3,716,171 (68.6) |
Quantitative |
64,780 (28.9) |
1,703,088 (31.4) |
Missing |
0 (0.0) |
1 (0.0) |
With regard to interval CRC after some period of time from colonoscopy, to assess the number of interval cancers as you mentioned, we need to know the date and results of colonoscopy. However, the accurate interval CRC rate based on colonoscopy could not be analyzed in this study because information on the date and results of colonoscopy was insufficient in the NHIS-NHID. In our study, interval cancer rate was defined as the number of subjects diagnosed with CRC within 1 year after FIT per 1000 subjects with negative FIT results as one previous study analyzed with National Cancer Screening Program (NCSP) data in Korea (Am J Gastroenterol 2018;113:611-21). In other words, in our study, interval cancer refers to cancer that was missed on a FIT, not a colonoscopy. We have added the following limitation in the Discussion section: “Third, since information on the date and results of colonoscopy was insufficient in the NHIS-NHID, the cancer rate missed by colonoscopy could not be analyzed.”
Other comments:
Abstract:
Results: I would suggest to report the number of interval CRCs detected in both study groups
Reply: To assess the number of interval cancers as you mentioned, we need to know the date and results of colonoscopy. In South Korea, examinees are undergoing colonoscopy in three ways. The first way is to undergo a colonoscopy as a secondary examination after FIT through the NCSP, the second way is to undergo a colonoscopy under medical insurance after visiting doctor, and the third way is to undergo a colonoscopy through a comprehensive health check-up regardless of medical insurance. When analyzing using the National Health Information Database (NHID), the date and results of colonoscopy are available in the case of the first way, while only the date of colonoscopy is available in the case of the second way. Moreover, neither the date nor the result of colonoscopy is available in the case of the third way. Therefore, the number of accurate interval CRC based on colonoscopy could not be analyzed in this study. In our study, interval cancer rate was defined as the number of subjects diagnosed with CRC within 1 year after FIT per 1000 subjects with negative FIT results as one previous study analyzed with National Cancer Screening Program (NCSP) data in Korea (Am J Gastroenterol 2018;113:611-21). In other words, in our study, interval cancer refers to cancer that was missed on a FIT, not a colonoscopy. We have added the following limitation in the Discussion section: “Third, since information on the date and results of colonoscopy was insufficient in the NHIS-NHID, the cancer rate missed by colonoscopy could not be analyzed.”
Methods:
Page 2, Study population section: the authors should provide more detailed description of the screening process in Korea. Age range for individuals with and without family history of CRC should be reported.
Reply: We provided more detailed description of the screening process in South Korea in the Method section, as follows. “The Korea National Cancer Screening Program (NCSP) provides a single annual FIT as an initial CRC screening examination for all Koreans over 50 years and those with positive FIT are offered a colonoscopy as a second examination. Participants were notified of the FIT results (reported as ‘positive’ or ‘negative’) by mail within 15 days. Also, participants who had a positive FIT result were contacted by telephone by the medical staff, informed of the positive result, and offered an appointment date for follow-up colonoscopy. All of these examinations were performed free of charge at a clinic or hospital designated as a CRC screening unit by the National Health Insurance Corporation (NHIC). To be designated as a CRC screening unit, a clinic or hospital must have colonoscopy equipment and at least one fulltime medical doctor. The population for the current study comprised men and women aged 50 years or older who received FITs through the NCSP between January 1, 2009, and December 31, 2011.”
In addition, we added the age range for individuals with and without family history of CRC as follows in the Results section: “The age ranges for those with and without family history of CRC were 50–100 years and 50–108 years, respectively.”
Description of the study population should also include how the history of CRC is determined (i.e. how information on family history is obtained)
Reply: We added how the history of CRC was determined as follows: The history of cancer including CRC (C, D01.0–D01.3) and IBD (K50, K51) was confirmed through the presence of the corresponding diagnostic code before FIT.”
Also, We described in detail how to obtain information about family history in the Method section: “Data on family history of CRC were collected on the basis of replies to a self-administered questionnaire. To receive the FIT provided by the NCSP, examinees are obliged to fill out a questionnaire that includes questions regarding family history of CRC. Subjects were recorded as having a family history of CRC if they had any FDRs (parents, siblings, or children) who had been diagnosed with CRC, regardless of age at diagnosis.”
A description of how the individuals eligible for screening are identified should be included.
Reply: Eligible subjects for CRC screening were all Koreans over the age of 50 without any special restrictions. The Korea National Cancer Screening Program (NCSP) provides an annual FIT as a CRC screening examination for all Koreans over 50 years free of charge without any special restrictions. The NHIS–NHID (the National Health Insurance Service-National Health Information Database) contains information on all screenees who participated in FIT screening through the NCSP. However, as shown in the table below, the FIT screening participation rate is low, ranging from 27-33%. Since our study reflects the results of only the screenees who participated in FIT screening, there is inevitably some selection bias.
<CRC Screening participation rates for NCSP in Korea, 2009-2011>
(Ref: Cancer Res. Treat.2017,49,798–806)
Variable |
2009 |
2010 |
2011 |
No. of invitations |
8,483,437 |
9,075,852 |
9,271,231 |
No. of participants |
2,281,444 |
2,794,663 |
3,049,112 |
Screening rate (%) |
26.9 |
30.8 |
32.9 |
We added this limitation in the Discussion section as follows: “Sixth, because only subjects who participated in FIT screening were analyzed, there may be some degree of selection bias.”
Does The Korea National Cancer Screening Program keep data in a central database?
Reply: Yes. The National Cancer Screening Program keeps the data in a central database (the National Health Information Database [NHID].
Page 2, lines 72-79: the numbers reported by the authors should be moved to the results section
Reply: As the reviewer recommended, we moved the numbers to the Results section.
Page 2, line 79 – please specify if the reported final number of individuals in the study population reflects the number of individuals who underwent FIT in the study period (although it is clearly shown on the Figure 1, it would be valuable for a reader to report it in the text)
Reply: We described the following sentences in the Results section: “Ultimately, a total of 5,643,438 participants who underwent FIT in the study period were included in the analysis.” “Of the 5,643,438 participants, 224,178 (3.97%) had a family history of CRC.”
Page 3, lines 87-97 – the authors report 4 different cut-off levels of haemoglobin in stool for a positive results ranging from 30 ng/ml to 100 ng/ml. The distribution of the cut-off levels among study population should be reported and compared between individuals with and without family history for CRC, as this significantly influences the interpretation of the results
Reply: As the reviewer pointed out, we investigated FIT analysis methods in more detail. As a result, qualitative and quantitative analysis accounted for 71.1% and 28.9%, respectively, in participants with family history of CRC, while qualitative and quantitative analysis accounted for 68.6% and 31.4%, respectively, in those without family history of CRC. In the data source (National Health Information Database [NHID] of the National Health Insurance Service [NHIS]) we analyzed, the brand name of each FIT kit was not included and the cutoff level was also not included in the case of qualitative analysis. Because qualitative analyses without cutoff level information occupied a very large proportion of study subjects and uncertain cutoff levels existed in some cases with quantitative analysis, we could not compare the distribution of the cutoff levels according to family history. We have added this limitation in the Discussion section as follows: “Fifth, the brand name of FIT kit was not included in the NHIS-NHID and the cutoff level was not included in the case of qualitative analysis. In addition, uncertain cutoff levels existed in some cases with quantitative analysis. Accordingly, we could not compare the cutoff level of FIT according to family history.”
|
Family History of CRC N=224,178 |
No Family History of CRC N = 5,419,260 |
Qualitative |
159,398 (71.1) |
3,716,171 (68.6) |
Quantitative |
64,780 (28.9) |
1,703,088 (31.4) |
Missing |
0 (0.0) |
1 (0.0) |
The rate of 3.97% for having a positive family history of CRC is relatively low.
There was a significant difference between the proportions of individuals who underwent colonoscopy in the last 5 years in the study groups. This should be discussed in the discussion section as a potential biasing factor lowering the positivity rate and diagnostic yield among individuals with a positive family history. Both individuals with and without family history of CRC who underwent colonoscopy within the last 5 years had significantly lower proportion of positive test results.
Reply: Thank you for your comments. We added the following sentences in the Discussion section: “Seventh, there was a significant difference in the history of colonoscopy within past 5 years between individuals without and with a family history of CRC. A higher percentage of colonoscopy history in individuals with a family history than in those without a family history may have acted as a potential biasing factor in lowering the positive rate and diagnostic yield of FIT in those with a family history.”
The authors should also discuss the possibility that interval cancers are a consequence of inadequate colonoscopy quality performed as a workup for a positive FIT. The proportion of interval cancers diagnosed after negative and positive FITs should be reported. The data on colonoscopy quality requirements in Korea should also be reported.
Reply: Thank you for your important comments. As mentioned in the earlier reply, in this study using the Korea National Health Information Database (NHID), information on the date and results of colonoscopy was insufficient. We have added this limitation in the Discussion section: “Third, since information on the date and results of colonoscopy was insufficient in the NHIS-NHID, the cancer rate missed by colonoscopy could not be analyzed.”
With respect to colonoscopy quality, NHID do not provide the detailed data on colonoscopy quality requirements. However, efforts have been made to improve the quality of colonoscopy in the NCSP in Korea. The Korean Society of Gastrointestinal Endoscopy (KSGE) has developed the National Endoscopy Quality Improvement Program (NEQIP) to improve the colonoscopy quality of NCSP (Korean J Gastroenterol 2014;64:320-32.Gut Liver 2016;10:699-705). This program incorporates qualifications of endoscopists, quality improvement for instruments available at the endoscopy unit and endoscopic procedures (including sedation and reprocessing of endoscopes), as well as measurement of outcomes of endoscopy screening.
Page 7, lines 4-6 – the statement that ‘FIT specificity was lower for subjects with a family history of CRC than those without’ is too strong, as the significance of this difference is very low (93.8% vs. 94.3%, aRR 0.994 [95% CI 0.990–0.999]). This comment also applies to lines 9-10 on the same page.
Reply: Thank you for your comments. We deleted the two sentences regarding specificity.
The proportion of individuals who had previous screening with a positive result (i.e. a polyp which was removed) should be reported in two study groups.
Reply: We added the proportion of individuals with the history of polypectomy in the two study groups in Table 1.
The discussion lacks a limitations section. Limitations of this study should be discussed to provide the reader better insight to the presented results.
Reply: There is a paragraph describing the limitations of our study in the discussion section. We mentioned a total of 8 limitations and fully discussed the limitations of our study.
We would like to thank the reviewer for the invaluable comments and appreciate the efforts to review our manuscript.

Round 2
Reviewer 2 Report
Dear Editor,
The authors addressed all the comments which were included in my primary review. My two major concerns were satisfactory addressed. I would recommend the authors to include the presented data from the tables into a supplementary appendix as they may be valuable for the readers. I do not have any further comments regarding methodology of this study.Author Response
Comments and Suggestions for Authors
Dear Editor,
The authors addressed all the comments which were included in my primary review. My two major concerns were satisfactory addressed. I would recommend the authors to include the presented data from the tables into a supplementary appendix as they may be valuable for the readers. I do not have any further comments regarding methodology of this study.
Reply: We would like to thank the reviewer for the invaluable comments and appreciate the efforts to review our manuscript.
As the reviewer recommended, we have added the following two tables as supplementary information. Also, we have added the following sentences in the manuscript.
“With regard to FIT analysis methods, the qualitative and quantitative analysis accounted for 71.1% and 28.9%, respectively, in participants with family history of CRC, while qualitative and quantitative analysis accounted for 68.6% and 31.4%, respectively, in those without family history of CRC (Table S1).” (Results section)
“During the study period, the participation rate of CRC screening (FIT) was low, ranging from 27-33% (Table S2) [23].” (Discussion section)
Supplementary information
Table S1. The proportion of qualitative and quantitative FITs
|
Family History of CRC N=224,178 |
No Family History of CRC N = 5,419,260 |
Qualitative FITs |
159,398 (71.1%) |
3,716,171 (68.6%) |
Quantitative FITs |
64,780 (28.9%) |
1,703,088 (31.4%) |
Missing data |
0 (0.0%) |
1 (0.0%) |
FIT, fecal immunochemical test
Table S2. Participation rate of colorectal cancer screening (FIT) in Korea, 2009-2011
|
2009 |
2010 |
2011 |
No. of invitations |
8,483,437 |
9,075,852 |
9,271,231 |
No. of participants |
2,281,444 |
2,794,663 |
3,049,112 |
Screening rate |
26.9% |
30.8% |
32.9% |
FIT, fecal immunochemical test
The above numbers are based on a previous paper (reference No. 23).

This manuscript is a resubmission of an earlier submission. The following is a list of the peer review reports and author responses from that submission.
Round 1
Reviewer 1 Report
Comments
In the Discussion, you correctly state that the different brands of FIT device and the different thresholds for positivity used in different areas of the Country represent a limit of the study. However, a description of both items (brands and positivity thresholds) should be reported in the methods.
Overall, the sensitivity of the screening program for CRC was 64%. This result is very low as compared with the literature (see for instance PMID:24658694) and it should be carefully considered when interpreting the study results. What are the possible explanations for this result? That’s why it is relevant to report the positivity thresholds used; further, are quality controls regularly carried out in the Labs involved in the screening program? Anyway, the study results are hardly generalizable to other Countries where higher sensitivity of FIT has been reported. This needs to be discussed.
Absolute numbers about the study outcomes need to be reported in the results session. In the present version, only relative data are shown, which limits the possibility to fully comprehend the results of the study and to interpret them. I suggest to add to table 1 some lines with the absolute numbers of positive FITs, screen-detected CRC, Interval Cancers, as well as positivity rates, screen-detected CRC rates and Interval Cancers rates.
A significant increase in Interval Cancer Rate among subjects with a family history of CRC cannot be automatically translated into an indication towards a more sensitive screening test (i.e., colonoscopy). Such result must be coupled with the absolute size of the problem. That is: what was the IC rate among subjects with a family history of CRC? Should it be one per 100,000, I would not push towards colonoscopy… should it be one per 1,000, than colonoscopy could be considered. That’s why it is necessary to show absolute numbers.
You refer to screenees as “patients” (abstract, results second line, …). In fact, fortunately undergoing a FIT does not make one become a patient… please replace “patients” with a more appropriate term, i.e., subjects, or persons.
Was the study approved by an Ethics Committee? Please add an ethics statement.